# Nutrient Intake and Menopausal Symptoms in Perimenopausal Women

**DOI:** 10.3390/nu17243887

**Published:** 2025-12-12

**Authors:** Miriam Bernadette Murphy, Geraldine Cuskelly, Patricia Heavey

**Affiliations:** SHE (Sport, Health & Exercise) Research Centre, Technological University of the Shannon, N37 HD68 Athlone, Irelandgeraldine.cuskelly@tus.ie (G.C.)

**Keywords:** menopausal symptoms, perimenopause, diet, nutrients

## Abstract

Background/Objective: Perimenopause, the final years of a woman’s reproductive life, may be symptomatic, affecting health and wellbeing. Lifestyle factors, such as diet, may alleviate these symptoms and improve health. Understanding nutrient intake at this stage of life is critical for tailoring dietary recommendations. This study investigated the nutrient intake of perimenopausal women in relation to adult guidelines and explored the associations between nutrients and symptoms. Methods: Irish perimenopausal women were recruited (>40 years old). Participants reported symptom severity via the Menopause Rating Scale (MRS). They completed a 4-day food diary along with a physical activity (PA) and demographic questionnaire. Hierarchal multiple regression was used to test the relationship between MRS and nutrients. A one-sample *t*-test or Wilcoxon signed-rank test was used to compare nutrient intake to the guidelines. Results: Participants (*n* = 63) had a mean age of 47.8 ± 3.5 years, a body mass index (BMI) of 27.0 ± 3.9 kg/m^2^ and an overall MRS score of 13.5 ± 6.9. Participants were not meeting dietary reference values (DRVs) for most nutrients, with intakes significantly below guidelines for fibre (*p* < 0.001), total vitamin D (*p* = 0.031), calcium (*p* < 0.001), and iron (*p* < 0.001), and exceeding guidelines for saturated fat (*p* < 0.001). Hierarchal multiple regression models (adjusted for BMI, age, marital status, and PA) showed that Vitamin E intake was a predictor of a higher urogenital sub-score (β = 0.405, *p* < 0.001). Conclusions: There was variable adherence amongst midlife women to dietary guidelines, which may impact on both current and future health status. Public health initiatives are required to address this.

## 1. Introduction

The permanent cessation of ovulation and menses, known as menopause, marks the end of reproductive capacity in females. The time preceding this, characterised by a reduction in circulating oestrogen and progesterone, irregular menstrual periods, and a range of symptoms, is known as perimenopause [1]. A host of symptoms have been attributed to this transitional life stage and are mostly categorised as either somatic (hot flushes, night sweats, joint and muscular discomfort), urogenital (bladder problems, vaginal dryness, sexual problems), or psychological (depressive mood, irritability, anxiety) [2]. In Ireland, the average age for menopause is 51 years [3], with symptoms commencing up to 7 years prior and often continuing into the postmenopausal stage. A recent Irish study reported that 78% of women in peri- or post- menopause describe symptoms as interfering with day-to-day activities [4].

Most women will experience symptoms of varying severity due to a decline in ovarian function. There are, however, a variety of symptom management strategies available, with Menopausal Hormonal Therapy (MHT) generally considered to be an effective first-line approach [5]. This treatment is, however, contraindicated for some individuals and may not be the first choice for others [6]. Some women may favour other approaches to complement MHT, including diet, physical activity, and other lifestyle strategies [7].

The literature on diet and menopausal symptoms is heterogeneous and inconclusive. A recent systematic review suggested that the consumption of foods high in sugar, saturated fat, ultra-processed foods, and caffeine may be associated with more severe menopausal symptoms, while a greater intake of vegetables, wholegrains, and unrefined foods may be linked to symptoms that are less severe [8]. Limited evidence has also linked certain key vitamins and minerals to improved symptoms, including vitamin E with vasomotor symptoms (VMS) such as hot flushes and night sweats [9,10] and zinc with memory [9].

In addition to ameliorating menopausal symptoms [11], adherence to a healthy diet (opting for lower-fat, higher-fibre foods [11,12,13], and reducing intake of animal products [11,12] has been suggested as a useful strategy for controlling body weight [12] and improving quality of life [13]. It has also been suggested that having a high BMI and central adiposity may be positively correlated with a more symptomatic perimenopause [14,15]. Midlife obesity is also associated with reduced quality of life [16] and an increased chronic disease risk [17,18]. Menopause is also associated with significant metabolic changes and a shift towards a more atherogenic lipid profile [19,20]. Due to a variety of factors, including a desire to maintain or lose weight at this stage of life, menopause has been labelled as a ‘window of opportunity’ [21] for initiating lifestyle changes, such as healthier dietary choices, when women may be motivated to alter their behaviours.

Adherence to healthy eating guidelines may also reduce the risk of certain diseases in later life, including cardiovascular disease (CVD), metabolic syndrome, diabetes, and some forms of cancer [22]. There are few studies investigating adherence by women to dietary recommendations during perimenopause. A recent systematic review of adult populations worldwide (including males and females) suggests that Ireland, alongside other high-income countries, may not be meeting national dietary recommendations, with consumption of grains and dairy generally lower than recommendations and intake of meat exceeding guidelines [23]. However, these findings are reflective of entire populations and do not focus on specific subgroups. Given the sparsity of literature, it would be useful to gain a better understanding of the compliance of perimenopausal women in Ireland with dietary recommendations.

The aim of this study was to investigate the diets of perimenopausal women living in Ireland by comparison to adult guidelines and to explore the association between diet and symptom severity.

## 2. Materials and Methods

### 2.1. Study Design

The study was cross-sectional and observational. A variety of methods (emails/social media posts/posters within work and community settings) were used to recruit a convenience sample. Participants were required to be resident in Ireland, >40 years old, and to self-identify as perimenopausal. Descriptions of the stages of menopause were based on the STRAW +10 criterion [24]. Criteria for inclusion were as follows: (i) experiencing menopausal symptoms alongside regular monthly periods of varying length or flow (early-stage perimenopause), or (ii) having experienced more than 3 months of amenorrhea, but not gone a full year without menses (late-stage perimenopause). Women who were pregnant, breastfeeding, or those with an early or surgical menopause were excluded due to their altered dietary requirements or recommendations. After being issued an information sheet and providing consent, participants completed an electronic screening questionnaire.

A priori power analysis was conducted to estimate the sample size needed for multiple regression modelling. Cohen’s approach to statistical power analysis was applied [25]. Assuming a medium effect size (ƒ2 = 0.15), a significance level of 0.05, and a desired power of 0.80, with 6 predictors in the model, a minimum of 60 participants was required.

Ethical approval for this study was granted by the Research Ethics Committee of the Technological University of the Shannon in November 2022. All data collected throughout the study was stored in accordance with GDPR EU 2016/679, and those involved were free to withdraw at any point up until the publication of the results. Participants who met the eligibility criteria were invited to attend to have a range of anthropometric measures taken and to report habitual dietary and physical activity data. A range of demographic data (age, education level, etc.) was also collected using an online form.

### 2.2. Data Collection

#### 2.2.1. Food Diaries

Participants completed a prospective, 4-day, semi-quantitative food record [26] including 1 weekend day, as this has been shown to provide a reliable estimate of intakes of key macro- and micronutrients in midlife women [27]. Quantitative analysis was completed using Nutritics software (Nutritics. (2019). Research Edition (v5.09*) [Computer software]. Dublin, Ireland). For the purpose of rating compliance to dietary guidelines, data were compared to published DRVs by the European Food Safety Authority (EFSA) whenever possible [28,29]. Guidelines from the World Health Organisation (WHO) [30,31] and the Health Service Executive (HSE) [32] were consulted for a small number of dietary components when specific EFSA recommendations were unavailable.

#### 2.2.2. Anthropometric Measurements

Height and weight (using a Seca 213 stadiometer (Seca, Hamburg, Germany) and Tanita MC-580 scales, (Tanita, Amsterdam, The Netherlands)) were recorded by a researcher trained in anthropometry. Participants were asked to abstain from food and drink for a minimum of 4 h before measurements. Additionally, they were requested to remove shoes, jewellery, and wear light clothing before using the scales.

#### 2.2.3. Physical Activity Questionnaire

The short-form version of the validated International Physical Activity Questionnaire (IPAQ) [33] was completed electronically. The number of MET (metabolic equivalent) minutes spent in low, moderate, and vigorous PA was scored based on the protocol by Forde [34], in addition to sedentary time. These values were converted to provide a physical activity level (PAL) value using the Institute of Medicine [35] ranges available in the EFSA protocol [36]. The estimated energy requirement (EER) was calculated as the product of (i) the estimated basal metabolic rate (BMR) (using Henry Oxford equation) [37] and (ii) PAL.

#### 2.2.4. Menopause Rating Scale (MRS)

Symptom severity was ranked using the MRS [38]. This previously validated 11-item questionnaire allows participants to self-report (using a Likert scale) between 0 (no symptoms) to 4 (very severe symptoms) for each menopausal symptom within the psychological, somatic, and urogenital domains. This provided an overall symptom score (out of 44), in addition to a sub-score for each symptom.

Previously published ranges based on the MRS categorise overall symptoms and sub-scores as no or little/mild/moderate/severe [39,40]. Overall MRS rankings of 0–4, 5–8, 9–16, and ≥17 are indicative of no or little, mild, moderate, or severe scores, respectively. For psychological sub-score, rankings of 0–1 are classified as no or little symptoms, 2–3 as mild, 4–6 as moderate, and ≥7 as severe. Somatic sub-score rankings of 0–2 indicate no or little symptoms, 3–4 are mild, 5–8 are moderate, and ≥9 are severe. Urogenital sub-score rankings of 0/1/2–3 and ≥4 are indicative of no or little/mild/moderate and severe symptoms, respectively.

#### 2.2.5. Mis-Reporters of Dietary Intake

A disparity has previously been observed between reported and actual dietary intake in midlife females [41]. For robustness, mis-reporters and plausible reporters of habitual intake were identified using the revised Goldberg cut-off [42]. The EFSA protocol [36] was applied to calculate a ratio of reported energy intake (estimated from food diaries) compared to BMR for an individual [37]. Lower cut-off points were <0.872, <0.996, and <1.12 for low, moderate, and high PA, respectively, while upper cut-off points of >2.249, >2.57, and >2.892 were applied for the same PA categories. Any participants outside the cut-off ranges were categorised as either under- or over-reporters and were excluded from the final data analysis.

#### 2.2.6. Statistical Analysis

Statistical testing was completed using the IBM SPSS statistics version 29 software. Dietary intakes were compared to DRVs using a one-sample *t*-test if data were normally distributed or a Wilcoxon signed-rank test for data that violated the assumption of normality. The median (interquartile range [IQR]) has been reported for data regarded as not normally distributed, whilst the mean (±standard deviation [SD]) was used to present data meeting assumptions of normality. Several predictive models were developed using hierarchical multiple regression. Age, BMI, physical activity, and marital status were included as covariates in regression models, as they have been shown previously to be associated with menopausal symptom severity [15,43,44,45]. For all tests, *p* < 0.05 was considered significant.

## 3. Results

### 3.1. General Characteristics of Participants

Seventy-seven women completed the study. After under-reporters of energy intake were excluded, 82% of participants (*n* = 63) were categorised as plausible reporters. There were no over-reporters. This cohort was predominantly married or cohabiting (84%), non-smokers (95%), with a third-level education (90%), in early-stage perimenopause (68%), with an average age of 47.8 ± 3.5 years. Their mean BMI was 27.0 ± 3.9 kg/m^2,^ with 35% within the healthy range, 41% living with overweight, and 24% living with obesity. In total, 21 percent were classified as having low activity levels, 52% as being moderately active, and 27% as being highly active (Table 1).

### 3.2. Perimenopausal Symptoms

Overall symptom score and domain sub-scores are presented in Table 2. The mean overall symptom score was 13.5 ± 6.9, which is indicative of moderate symptoms, with moderate or severe symptoms present for 81% of the cohort. Psychological sub-score was classified as severe for 49% of participants, while 45% were deemed to have a severe urogenital sub-score. Somatic symptoms were less severe overall amongst the group, with only 5% having a sub-score classified as severe in this domain. The prevalence of individual symptoms is shown in Table 3. The most commonly reported symptoms were irritability (86%) and physical and mental exhaustion (84%), whereas the least commonly reported symptoms were vaginal dryness (54%), followed by heart discomfort (57%).

### 3.3. Macronutrient Intakes

Median daily energy intake was 1805.0 kcal (IQR 408.0), with a minimum of 1283.3 kcal and a maximum of 3037.9 kcal (Table 4). Median daily intakes of protein and fat were 79.0 g (IQR 26.0) and 71.9 g (IQR 30.5), respectively. Mean carbohydrate intake was 193.2 g (±51.6), median intake of dietary fibre, 19.2 g (IQR 7.0), and 4.0% (IQR 3.4) of total energy from free sugars. In terms of macronutrients as a percentage of total energy intake, 17.9% was provided by protein, 36.9% from fat, and 42.8% from carbohydrate sources. Saturated fat provided a mean of 13.5% (±3.5) of total energy intake, whilst the median intake of omega-3 was 702.8 mg (IQR 831.0). Alcohol intake ranged between 0 and 47.8 g per day, with the median intake equivalent to 0.4 standard drinks per day (IQR 1.2) (Table 4).

### 3.4. Micronutrient Intakes

Median calcium intake was 649.6 mg/day (IQR 381.9) (Table 5). The median daily intake of key vitamins, A, D, E, and K, was 843.2 μg (IQR 851.6), 5.3 μg (IQR 9.1), 8.0 mg (IQR 13.8), and 32.0 μg (IQR 47.4, respectively, whilst daily vitamin C intake was 111.8 mg (IQR 122.8) and intake of vitamin B12 was 4.3 μg (IQR 5.5). Median intake of folic acid was 200.0 μg (IQR 318.6) per day, with 52% of the cohort consuming folic acid in the form of fortified foods/supplements. Iron, potassium, and sodium consumption were 8.6 mg (IQR 5.9), 2254.2 mg (IQR 1005.5), and 1.9 g (IQR 0.8), respectively (Table 5). The median daily intake of caffeine was 44.8 mg (IQR 75.3), with intakes ranging between 0 and 275.8 mg.

### 3.5. Adherence to Dietary Guidelines

Nutrient intakes in comparison to DRVs are presented in Table 4 and Table 5. Out of the dietary guidelines explored, six recommendations were met by ≥80% of participants (protein, eicosapentaenoic acid [EPA] and docosahexaenoic acid [DHA], niacin, thiamine, caffeine, and alcohol).

Twenty-seven percent of the sample were consuming adequate energy to meet their individualised target based on their age, estimated resting energy expenditure (REE), and reported PAL. Reported energy intake for the group was significantly lower than estimated requirements (*p* < 0.001, r = 0.83), with 73% of participants falling below this calculated value; an average deficit of 238.9 kcal (IQR 511.0).

Based upon EFSA recommendations for macronutrients, 87.3% of participants met protein guidelines of >0.83 g per kg body weight. Carbohydrate guidelines (45–60% of total energy) were met by 34.9%. The remainder of the cohort fell below this range, and no participant exceeded the upper range of >60% of total energy from carbohydrates. Recommendations that <30% of total energy be derived from fat were met by 28.6% of the cohort. Participants exceeded this recommendation by a mean of 1.9% of total energy (*p* = 0.003, r = 0.2). The majority were consuming more saturated fat than the guideline of <10% of overall energy (*p* < 0.001, r = 0.78), with only 11.1% falling below this target. Dietary fibre intakes were generally below guidelines (*p* < 0.001, r = 0.28), with 17.5% of participants achieving the recommendation of >25 g per day.

The percentage of women meeting the DRV for vitamins and minerals ranged from 7.9 to 100% (Table 5). Calcium intake was significantly lower than the DRV (*p* < 0.001, r = 0.55), with the majority of women (76.2%) falling below 950 mg per day. Only a small number of participants were meeting DRVs for iron, selenium, and potassium, with the guideline amounts met by 17.5%, 9.5%, and 7.9% of the sample, respectively (*p* < 0.001 in all cases, r = 0.61, 0.55, 0.81, respectively). Median intake of vitamin A (843.2 μg RE/day, IQR 851.6) was above the guideline (*p* = 0.002, r = −0.38), with 57.1% meeting the recommended amount. Although participants’ vitamin C and E intake did not meet DRVs (59.7% and 66.6% respectively), the difference was not significant compared to the DRV (*p* = 0.215 and *p* = 0.529 respectively). Intakes of vitamins D and K were significantly less than the guidelines (*p* = 0.031, r = 0.27, and *p* < 0.001, r = 0.51, respectively), with only 22.2% and 19.0% meeting the recommended amount. Just over half of women (50.8%) were meeting the DRV for vitamin B12, and intake was not significantly different from the DRV (*p* = 0.385). Almost a quarter of participants (23.8%) included ≥200 mg of folic acid in the form of fortified foods or supplements, with intake significantly less than the DRV for the entire cohort (*p* < 0.001, r = 0.45); however, intake of total dietary folates was not significantly different from the guideline (*p* = 0.114). Mean sodium consumption exceeded the national target amount of ≤1.6 g/d (*p* < 0.001, r = −0.64), with only 1 in 4 participants (23.8%) consuming below this national target.

Assuming projected alcohol intake for each participant remained constant throughout the week, 82.5% of women would have met the national guideline of ≤11 units per week [32]. In terms of caffeine, none of the cohort exceeded 400 mg per day [29].

### 3.6. Predictors of Perimenopausal Symptom Severity

A number of hierarchical multiple regression models were used to explore whether dietary components improved the prediction of overall and sub-score symptom severity compared to a group of demographic variables alone. Each model included the covariates of BMI, age, marital status, and PA, which were entered concurrently. Over 90% of participants were educated to the third level, with a similarly high percentage of non-smokers (95%). Neither of these variables was found to have a significant predictive effect on the models created and were therefore not included as covariates in the final models. Vitamin E was the only nutrient that emerged as a significant predictor, and only for the urogenital sub-score (Table 6). No other nutrients were identified as predictors of overall menopausal symptoms or sub-scores.

Models 1 and 2 investigated predictors of the urogenital sub-score. In model 1 the co-variates were entered, explaining 10.5% of the variance in the model (adjusted R^2^ = 0.105). In Model 2, Vitamin E was added, which made a significant improvement to the model (∆R^2^ < 0.001, *p* < 0.05), raising the total explained variance to 25.6% (adjusted R^2^ = 0.256). Vitamin E was the strongest predictor of urogenital sub-score (β = 0.405, *p* < 0.001), with age (β = 0.376, *p* = 0.002), and BMI (β = 0.248, *p* = 0.042) also emerging as significant predictors.

## 4. Discussion

To our knowledge, this is the first study exploring dietary intake and perimenopausal symptoms in women in Ireland. The results of this study indicate that adherence to dietary guidelines amongst this cohort was variable. This may have implications, with potential impacts on aspects of both midlife and future health status, including body composition, cardiovascular, bone, and gut health, and cognitive function.

Of the macronutrients explored, the highest levels of adherence to recommendations were achieved for protein (87.3% meeting the target). Adequate protein intake during perimenopause has been suggested to reduce the likelihood of excessive weight gain during this life stage [18]. However, the intake of saturated fat for this group (13.5% of total energy) exceeded the recommendation of ≤10% of total energy. This level of intake reflects the national average intake of saturated fat (between 13 and 14%), which has been consistently elevated in Ireland for nearly 20 years [48]. Women are at an increased risk of CVD, commencing during the perimenopausal stage [49] and continuing post menopause [50]. High saturated fat intake is a potential risk factor in CVD due to its role in increasing low-density lipoprotein cholesterol (LDL-C) levels, and in promoting both atherogenesis and inflammation [51]. Evidence from observational studies and randomised controlled trials (RCTs) has suggested that lower intake of saturated fats, and, in particular, the substitution of these fats for polyunsaturated fatty acids, is associated with improved cardiovascular outcomes [51]. A systematic review of 19 studies reported that a small body of research has found that a higher saturated fat intake is associated with a more symptomatic menopausal stage [8]; however, these findings were not corroborated by the current study. However, the design, size, and heterogeneity of the studies included in the systematic review mean that the evidence for this association is still equivocal.

Existing research suggests that menopause may impact cognition and dementia risk in women [52]. Consumption of certain dietary components, particularly pro-inflammatory foods, such as those high in saturated fat, may adversely affect cognitive function in postmenopausal women [53]. However, a systematic review has highlighted the limited and inconclusive nature of research in this area [54]. Although some studies show a positive relationship between saturated fat intake and risk of cognitive disorders, others suggest an inverse association [55]. Nevertheless, as the women in the current study were consuming saturated fat at levels largely exceeding guidelines, this may be of concern for future cognitive health risks.

Thirty-five percent of the current cohort consumed carbohydrates at a level within the recommended range of 45–60% of total energy intake, with no one exceeding 60%. In the current study, carbohydrate intake was not a predictor of menopausal symptoms. A higher carbohydrate quality index has previously been shown to be inversely associated with both somatic and psychological symptoms; however, this was only in Iranian postmenopausal women [56]. Participants of our study had low adherence to recommendations for dietary fibre, consuming about 20.6 g per day. Menopause has been linked to potential negative effects on gut health, with sex hormones and ageing both playing roles in the microbiome [57]. Ovarian ageing is associated with less diversity in gut microbiota [58], which may be ameliorated via modifiable risk factors such as a diet including sufficient fibre [57]. In addition to improved intestinal outcomes, adequate dietary fibre may be protective against other complications, including CVD, obesity, and metabolic disorders [59]. Additionally, findings from The Study of Women’s Health Across the Nation (SWAN), a large cohort of middle-aged women (*n* = 3054), suggest that dietary fibre is inversely associated with depressive symptoms in the premenopausal stage; however, this effect does not appear to continue into perimenopause [60]. Fibre was not a significant predictor of perimenopausal symptom severity in the current study. Conversely, an Iranian study including 393 postmenopausal women did find an improvement in VMS, with higher dietary fibre intakes, when measured as an element of a carbohydrate quality index [56]. Fruit and vegetable servings were not directly quantified in the current study; however, only 41% were meeting the DRV for vitamin C, 8% were meeting potassium targets, and 17% of the women were consuming >25 g of dietary fibre, which may all be potential indicators of low fruit and vegetable intake. A Mediterranean-style diet, rich in fruits and vegetables, has previously been recommended during perimenopause by the European Menopause and Andropause Society (EMAS) to improve both current and future health. Mediterranean dietary patterns have been suggested to reduce the risk of CVD and certain cancers, maintain bone density and cognitive function, and reduce VMS and depressive symptoms associated with menopause [61].

Caffeine and alcohol intake have both been associated with more severe symptoms during midlife in observational studies [8,62]. Previous research in Slovakian midlife women has shown that alcohol consumption is associated with vaginal dryness and night sweats [45]. Adherence to recommended limits for these nutrients was high amongst the current cohort, although a positive association between these dietary components and symptom severity was not observed. However, aiming to achieve alcohol guidelines may improve other health outcomes, given alcohol’s established role as a modifiable risk factor for various morbidities, including cancer [63].

The decline in oestrogen associated with ovarian ageing has been linked to an increase in the demineralisation of bones, putting women at greater risk of osteopenia and osteoporosis after menopause [64]. Perimenopause has been suggested as an opportune time to intervene in order to prevent degenerative bone diseases [65]. Less than a quarter of this cohort were compliant with advice to include 950 mg of calcium on a daily basis, potentially impacting their risk of osteoporosis and fractures in later life [65]. The current study did not identify a positive association between calcium intake and menopausal symptom severity. Similarly, a large-scale study of 23,020 participants taking part in the US-based Women’s Health Initiative (WHI) found that a combined calcium and vitamin D supplement had no effect on the severity of symptoms recorded using a study-specific questionnaire during the postmenopausal stage [66]. Previous narrative reviews have summarised the effect of vitamin D on midlife health and climacteric symptoms, highlighting the link between deficiency and adverse health outcomes, including CVD, diabetes, depression, suboptimal cognitive function [67], and urogenital and other menopausal symptoms [68]. Within this cohort, a majority (78%) of participants were not adhering to the EFSA advice to supplement year-round with 15 µg of vitamin D per day, comparable to a recent narrative review, which suggested that more than 90% of the Irish population are not meeting the recommended daily allowance of vitamin D [69]. Although seasonal variations in vitamin D were not accounted for in the current study, subcutaneous vitamin D synthesis is limited in Ireland due to its northern latitude (52–55° N) [69]. The importance of midlife sufficiency of vitamin D cannot be overstated, with as many as 86% of postmenopausal women living with suboptimal serum levels [70]. Supplementation has been encouraged as a useful strategy in preventative healthcare, with specific beneficial effects on bone health [71] and a potential role in reducing disorders of the cardiovascular system, improving metabolic markers, and enhancing mental health and cognition [67]. A recent North African cross-sectional study of 168 women suggested that there was no positive association between vitamin D intake and their overall MRS score or individual domain sub-score [72]. Previously, a study analysed data from a subset of 530 participants from across the US as part of the WHI trial. This observational study also found no significant association between vitamin D intake and severity of midlife symptoms once the data were corrected for multiple testing [66]. The age range of participants in the WHI and the Egyptian study was 51–80 years and 50–70 years, respectively, which indicates a largely postmenopausal cohort. A consensus statement from EMAS has advised against the supplementation of vitamin D as a treatment to ameliorate postmenopausal symptomology [73], with no specific mention of the perimenopausal stage. Studies exploring the effects of vitamin D on midlife health and symptom severity pose complex challenges due to a number of confounding factors. There is also a lack of high-quality RCT studies in this area [71].

Vitamin E has previously been found to alleviate postmenopausal symptom severity. A systematic review, including 16 studies and 1815 participants, found a reduction in both hot flushes and vaginal changes accompanied by an improved lipid profile following supplementation with vitamin E. Conversely, the current study found an unexpected association between a higher intake of vitamin E and an increased urogenital sub-score, with a statistical model including covariates accounting for 26% of the variance in symptom severity for this domain. However, this may be explained by the high rate of supplement use. Notably, 75% (*n* = 47) of participants were taking supplements containing varying amounts of vitamin E, including multivitamins, menopause-specific supplements, and evening primrose oil. Research has suggested an association between more severe menopausal symptoms and complementary and alternative medicine use [74]. Phytoestrogens and tocopherols are modulators of oestrogen receptors and have been suggested as a useful alternative to MHT in the management of menopausal symptoms [75]; however, there are no definitive guidelines on dosage.

The current study was limited by the use of a convenience sample. This resulted in a cohort with most participants (90%) having received a university-level education, disproportionate to the national average of 65% of women educated to a third level in Ireland [76], and, as such, the results cannot be generalised. Future research utilising a more diverse population (e.g., women with different educational levels, socio-economic status, ethnicity, etc.) is needed. The sample size was small, which may have resulted in an uneven spread of symptom severity categories, and, therefore, the results cannot be generalised. However, as an observational study, a causal effect was not being investigated, with results only applicable to this specific sample. The decision not to include MHT status in the statistical modelling may also have limited the study. A Cochrane review concluded that hormone replacement is effective in the treatment of perimenopausal symptoms [77]. Those taking MHT may have had some of their symptoms reduced or alleviated. However, some participants chose not to disclose their MHT status, and no information on dosing or the type of MHT was collected. More research exploring the interaction of MHT with dietary habits and symptom severity would shed light on this further. Strengths of this study include the use of a 4-day food diary, providing a wealth of information about dietary intake for a range of parameters. However, self-reported data introduces the potential for reporting bias, and estimated data is subject to errors which must be acknowledged [78]. Additionally, while DRVs are broad targets aimed at populations, they can overlook groups with distinct physiological or metabolic needs. More intervention studies investigating a whole diet approach are needed to better understand associations between diet and perimenopausal symptoms.

## 5. Conclusions

The uniqueness of this study, focusing exclusively on the perimenopausal stage, presents an interesting insight into an area that has been underexplored to date. This study provides useful information to help guide further research, ultimately contributing to policy and the development of recommendations and supports for midlife women. The promotion and communication of dietary guidelines for midlife females may also be beneficial from a public health perspective. This study reveals more about the diets of midlife women in Ireland, bringing to light the issue that many are not meeting DRVs. This needs to be addressed to protect current and future health.

## Figures and Tables

**Table 1 nutrients-17-03887-t001:** Participant characteristics.

Characteristic (*n* = 63)	Category	Mean ± SD	*n*	%*n*
Age (years)		47.8 ± 3.5		
Body composition measurement	BMI (kg/m^2^)	27.0 ± 3.9		
	Waist circumference (cm)	90.1 ± 10.5		
	Body fat (%)	32.4 ± 5.9		
BMI category	Healthy 18.5–24.9 (kg/m^2^)		22	35
	Overweight 25–29.9 (kg/m^2^)		26	41
	Obesity (≥30 kg/m^2^)		15	24
Marital status	Married/cohabiting		53	84
	Single or divorced		9	14
	Undisclosed		1	2
Education status	Not attained 3rd level		6	10
	Attained 3rd level		57	90
Smoking status	Smoker		3	5
	Non-smoker		60	95
Physical activity level	Low		13	21
	Moderate		33	52
	High		17	27
Perimenopausal stage	Early stage		43	68
	Late stage		20	32
MHT status	MHT		19	30
	No MHT		39	62
	Undisclosed		5	8

BMI Body mass index; MHT; menopausal hormone therapy.

**Table 2 nutrients-17-03887-t002:** Menopause Rating Scale (MRS) score and classification for overall symptoms and symptom sub-score domains.

Symptom Score/Sub-Score	Menopausal Rating Scale (MRS) Score	Menopause Rating Scale (MRS) Classification		
(*n* = 63)		None/Mild	Moderate	Severe
	(Mean ± SD)	*n* (%*n*)	*n* (%*n*)	*n* (%*n*)
Overall symptom score	13.6 ± 6.2	12 (19%)	34 (54%)	17 (27%)
Psychological sub-score	5.9 ± 3.1	17 (27%)	15 (24%)	31 (49%)
Somatic sub-score	4.5 ± 2.4	31 (49%)	29 (46%)	3 (5%)
Urogenital sub-score	3.2 ± 2.2	16 (25%)	19 (30%)	28 (45%)

**Table 3 nutrients-17-03887-t003:** Presence of individual symptoms of the Menopause Rating Scale (MRS).

Individual Symptom		Presence of Symptoms	
*(n* = 63)	Not Present		Present
	*n* (%*n*)		*n* (%*n*)
Irritability	9 (14%)		54 (86%)
Physical and mental exhaustion	10 (16%)		53 (84%)
Sleep problems	12 (19%)		51 (81%)
Depressive mood	14 (22%)		49 (78%)
Anxiety	15 (24%)		48 (76%)
Sexual problems	15 (24%)		48 (76%)
Joint and muscular discomfort	16 (25%)		47 (75%)
Bladder problems	24 (38%)		39 (62%)
Hot flushes	25 (40%)		38 (60%)
Heart discomfort	27 (43%)		36 (57%)
Dryness of vagina	29 (46%)		34 (54%)

**Table 4 nutrients-17-03887-t004:** Key nutrient intake in relation to guidelines.

Nutrient	Mean ± SD	Median (IQR)	DRV	Participants Meeting DRV Target/Range *n* (%*n*)	*p*-Value	Effect Size r
Energy kcal/d	1836.7 ± 350.9	1805.0 (408.0)	Individualised ^af^	17 (27.0)	<0.001 *	0.38
Protein						
Total, g/d	81.8 ± 19.1	79.0 (26.0)	0.83 ^a^	55 (87.3)	<0.001 *	−0.37
E%	18.1 ± 4.2	17.9 (4.7)				
Carbohydrate						
Total, g/d	193.2 ± 51.6	191.5 (70.2)				
E%	41.2 ± 7.7	42.8 (8.0)	45–60 ^a^	22 (34.9)	<0.001 *	0.43
g/kg of bodyweight/d	2.7 ± 0.8	2.8 (0.8)				
Free sugars (E%)	4.0 ± 2.5	4.0 (3.3)	≤5 ^b^	44 (69.8)	0.052	
Dietary fibre (g/d)	20.6 ± 7.2	19.2 (7.0)	≥25 ^a^	11 (17.5)	<0.001 *	0.28
Fat						
Total, g/d	76.6 ± 20.2	71.9 (30.5)				
E%	37.5 ± 6.6	36.9 (6.5)	20–35 ^a^	18 (28.6)	0.003 *	0.20
g/kg/d	1.1 ± 0.3	1.0 (0.4)				
SFA (E%)	13.5 ± 3.5	13.1 (4.3)	≤10 ^c^	7 (11.1)	<0.001 ^†^	0.78
EPA + DHA (mg/d)	1071.6 ± 951.6	702.8 (831.0)	250 ^a^	60 (95.2)	<0.001 *	−0.36
Alcohol (standard drinks/d)	0.7 ± 1.0	0.4 (1.2)	≤11 ^d^	52 (82.5)	<0.001 *	0.50
Caffeine (mg/d)	58.5 ± 64.5	44.8 (75.3)	≤400 ^e^	63 (100)	<0.001 *	0.53

d day; DHA docosahexaenoic acid; DRV dietary reference value; E% percentage of energy intake; *EPA* eicosapentanoic acid; SD standard deviation; SFA saturated fatty acids, (*n* = 63); ^a^ EFSA [28]; ^b^ WHO [30]; ^c^ WHO [31]; ^d^ HSE [32]; ^e^ EFSA [29]; ^f^ Individualised based on age and physical activity level (PAL) MJ/d (calculated using resting energy expenditure (REE) * PAL): 40–49 years PAL 1.4: 7.5 MJ; PAL 1.6: 8.6 MJ, PAL 1.8: 9.7 MJ; 50–59 years PAL 1.4: 7.5 MJ; PAL 1.6: 8.5 MJ, PAL 1.8: 9.6 MJ. ^†^
*p* < 0.05. *p*-value significance determined by a one-sample *t*-test or * Wilcoxon signed-rank test. Intake compared to DRV. Data compared to upper range (carbohydrate E%, fat E%), where appropriate.

**Table 5 nutrients-17-03887-t005:** Key micronutrient intake in relation to guidelines.

Micronutrient	Mean ± SD	Median (IQR)	DRV	Participants Meeting DRV Target/Range *n* (%*n*)	*p*-Value	Effect Size r
Vitamins						
Vitamin A (μg RE/d)	963.5 ± 668.0	843.2 (851.6)	650 ^a^	36 (57.1)	0.002 *	−0.38
Vitamin C (mg/d)	148.8 ± 169.4	111.8 (122.8)	95 ^a^	26 (41.3)	0.215	
Total vitamin D (μg/d)	18.2 ± 43.7	5.3 (9.1)	15 ^a^	14 (22.2)	0.031 *	0.27
Vitamin E (mg/d)	27.6 ± 101.1	8.0 (13.8)	11 ^a^	23 (36.5)	0.800	
Phylloquinone/Vitamin K (μg/d)	59.0 ± 87.9	32.0 (47.4)	70 ^a^	12 (19.0)	<0.001 *	0.51
Thiamin/B1 (mg/MJ)	3.0 ± 4.9	1.3 (1.3)	0.1 ^a^	63 (100)	<0.001 *	−0.87
Riboflavin/B2 (mg/d)	2.5 ± 2.6	1.4 (1.6)	1.6 ^a^	25 (39.7)	0.699	
Niacin/B3 (mg NE/MJ)	35.3 ± 16.9	32.2 (15.2)	1.6 ^a^	60 (95.2)	<0.001 *	−0.87
Vitamin B6 (mg/d)	3.7 ± 5.0	1.8 (2.3)	1.6 ^a^	38 (60.3)	0.018 *	−0.30
Folate (μg DFE/d)	364.5 ± 369.6	207.8 (295.8)	330 ^a^	26 (41.3)	0.114	
Folic acid (μg/d)	232.4 ± 212.7	200.0 (318.6)	200 ^b^	15 (23.8)	<0.001 *	0.45
Cobalamin/B12 (μg/d)	6.9 ± 4.5	4.3 (5.5)	4 ^a^	32 (50.8)	0.385	
*Minerals*						
Sodium (g/d)	2.1 ± 0.6	1.9 (0.8)	≤1.6 ^c^	15 (23.8)	<0.001 *	−0.64
Potassium (mg/d)	2390.2 ± 766.1	2254.2 (1005.5)	3500 ^a^	5 (7.9)	<0.001 *	0.81
Calcium (mg/d)	737.9 ± 339.4	649.6 (381.9)	950 ^a^	15 (23.8)	<0.001 *	0.55
Magnesium (mg/d)	285.2 ± 123.2	250.8 (120.6)	300 ^a^	20 (31.7)	0.31	
Iron (mg/d)	11.5 ± 7.9	8.6 (5.9)	16 ^a^	11 (17.5)	<0.001 *	0.61
Zinc (mg/d)	10.7 ± 8.7	8.0 (5.8)	7.5 ^ad^	35 (55.6)	0.044 *	−0.25
Copper (mg/d)	2.2 ± 4.8	1.1 (0.6)	1.3 ^a^	22 (34.9)	0.095	
Manganese (mg/d)	3.3 ± 1.7	2.9 (1.8)	3 ^a^	29 (46.0)	0.962	
Selenium (μg/d)	40.8 ± 22.5	34.8 (19.0)	70 ^a^	6 (9.5)	<0.001 *	0.55
Iodine (μg/d)	134.9 ± 115.7	102.4 (88.3)	150 ^a^	17 (27.0)	0.003 *	0.30

d day; DFE dietary folate equivalents; DRV dietary reference value; NE niacin equivalent; RE retinol equivalent SD standard deviation. EFSA [28] ^a^; FSAI [46] ^b^; FSAI [47] ^c^, (*n* = 63) ^d^ Phytate intake not recorded in current sample, DRV based on intake of 300 mg/d * *p* < 0.05. *p*-value significance determined by one-sample Wilcoxon signed-rank test. Intake compared to DRV.

**Table 6 nutrients-17-03887-t006:** Hierarchical multiple regression analyses for urogenital sub-score.

Urogenital Sub-Score				
	Model 1		Model 2	
	B (SE)	β	B (SE)	β
	R^2^ change = 0.046 *		R^2^ change <0.001 *	
Constant	−9.418 (4.315)		−13.118 (4.074)	
Age	0.176 (0.077)	0.283 *	0.234 (0.072)	0.376 *
Marital status	0.949 (0.780)	0.156	0.718 (0.714)	0.118
PAL (moderate)	0.975 (0.684)	0.227	0.794 (0.626)	0.185
PAL (high)	0.500 (0.771)	0.103	0.533 (0.704)	0.110
BMI	0.100 (0.071)	0.181	0.136 (0.065)	0.248 *
Vitamin E			0.009 (0.002)	0.405 *
Final model R^2^ _adj_			0.256 *	

* *p* < 0.05 B unstandardized regression coefficient; SE standard error of the coefficient; β standardised coefficient; PAL physical activity level; BMI body mass index; R^2^_adj_ Adjusted R-squared (*n* = 63).

## Data Availability

The datasets generated for this study are available from the corresponding author upon reasonable request.

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
