# Peer review of "Nutrient Intake and Menopausal Symptoms in Perimenopausal Women"

_nutrients, 2025, doi:10.3390/nu17243887_

Round 1
Reviewer 1 Report
Comments and Suggestions for Authors
This manuscript describes a well-conducted cross-sectional, observational study that adds to current understanding of the nutrient intake of perimenopausal women. A few minor points that need attention are outlined below:
- Within the introduction greater consistency could be used when describing how symptoms are categorized e.g. symptoms are classified as somatic, urogenital or psychological in lines 36-37 but later referred to as climacteric (line49) and vasomotor (line 52-53). It would be useful for the reader to explain some of these terms and/or provide examples of some of the associated symptoms
- Mean overall symptom score differs between text (line 174) and table 2. As does % deemed to have severe urogenital score (44% in text, line 176 and 45% in table)
- Consider adding carbohydrate as % energy intake to table 4 in line with other macronutrients and as described in text (line 231-232)
- Typo, line 209 ‘media’
- Typo, line 241’PRI’
- Results, section 3.6 (and table 6) describes multiple regression analysis for predictors of urogenital sub-score. It is not clear if predictors of the other domains (e.g. psychological) were also investigated. Please clarify.
Author Response
Comment 1: This manuscript describes a well-conducted cross-sectional, observational study that adds to current understanding of the nutrient intake of perimenopausal women. A few minor points that need attention are outlined below:
Response 1: Thank you for your comments. The relevant changes have been addressed and the manuscript is now hopefully at a publishable standard. The authors would like to thank the reviewers for their very helpful feedback.
Comment 2: Within the introduction greater consistency could be used when describing how symptoms are categorized e.g. symptoms are classified as somatic, urogenital or psychological in lines 36-37 but later referred to as climacteric (line49) and vasomotor (line 52-53). It would be useful for the reader to explain some of these terms and/or provide examples of some of the associated symptoms.
Response 3. Thank you. This has now been clarified and examples have been provided in line 36-38 under classification. Climacteric has now been replaced by menopausal for more consistency (L51). Examples of vasomotor symptoms (hot flushes and night sweats) have been added to line 55.
Comment 4: Mean overall symptom score differs between text (line 174) and table 2. As does % deemed to have severe urogenital score (44% in text, line 176 and 45% in table).
Response 4: Thank you for highlighting this. The text has been changed to 45% (the number had not been rounded up) (line 181)
Comment 5: Consider adding carbohydrate as % energy intake to table 4 in line with other macronutrients and as described in text (line 231-232).
Response 5: Thank you for this suggestion- they were omitted in error when pasting into the template. We have added these values to table 4.
Comment 6: Typo, line 209 ‘media’ Response 6: This has been replaced with ‘median’
Comment 7: Typo, line 241’PRI’ Response 7: This has been replaced with ‘DRV’
Comment 8: Results, section 3.6 (and table 6) describes multiple regression analysis for predictors of urogenital sub-score. It is not clear if predictors of the other domains (e.g. psychological) were also investigated. Please clarify.
Response 8: Yes, the other domains were also investigated but were not significant. We have amended the text to reflect this;
Line 264: Vitamin E was the only nutrient that emerged as a significant predictor and only for the urogenital sub-score (Table 6). No other nutrients were identified as predictors of overall menopausal symptoms or sub-scores.
and also amended the title of Table 6 to ‘Hierarchical multiple regression analyses for urogenital sub-score.
Reviewer 2 Report
Comments and Suggestions for Authors
The manuscript represents an important contribution to the field of nutrition and women’s health. The study provides valuable insight into the nutritional habits of perimenopausal women in Ireland and the relationships between nutrient intake and menopausal symptom severity. The methodology is appropriate, and the results are clearly presented. The manuscript is well-prepared and presents results of public health relevance. However, incorporation of the recommendations will enhance clarity, robustness, and impact of the work.
Recommendations and comments:
- In the introduction it is stated that “It has also been suggested that having a high BMI may be positively correlated with a more symptomatic perimenopause [14]. Midlife obesity is also associated with reduced quality of life [15] and an increased risk of chronic disease [16,17].” However, I consider it important to complement this information with current research showing that what appears to be particularly risky in midlife is metabolically unhealthy obesity, which is associated with metabolic abnormalities and, in addition to a less favourable lipid profile, is often accompanied by elevated levels of uric acid and liver enzymes—factors typically linked to metabolic syndrome (g., “Differences in body composition between metabolically healthy and unhealthy midlife women with respect to obesity status”). It is also worth mentioning research that have found abdominal obesity to negatively influence the occurrence of menopausal symptoms in women of European origin („Association of menopausal symptoms with obesity in Slovak women“).
- „Age, physical activity, and marital status were included as covariates in the regression models, as they have previously been shown to be associated with menopausal symptom severity [40,41].“ However, the cited evidence is drawn exclusively from Chinese populations. Given that menopausal symptoms are known to be strongly influenced by ethnic and population-specific factors, it would be important to incorporate findings from studies conducted in European-origin populations as well. These studies have demonstrated that physical activity, obesity (BMI > 30, WHR > 0.89), alcohol intake, and socio-demographic factors—such as marital status and place of residence (town vs. village)—significantly affect menopausal symptomatology in midlife women. For example, menopausal status, BMI, residence, and alcohol consumption were identified as significant predictors of vaginal dryness. Moreover, psychological midlife symptoms were found to be primarily associated with physical health, lack of physical activity, use of oral contraceptives, alcohol consumption, and smoking in previous research (e.g., “The importance of female reproductive history on self-reported sleep quality, mood, and urogenital symptoms in midlife”; “The reproductive parameters, lifestyle and health factors in relation to physiological menopausal symptoms in Slovak women”). Some of these findings—particularly those regarding the influence of alcohol on menopausal symptoms—would be valuable to address in the discussion.
- The association between higher dietary vitamin E intake and elevated urogenital symptom scores is unusual. The Discussion should further explore potential mechanisms, confounding factors, or alternative explanations.
- The overrepresentation of highly educated participants should be acknowledged as a limitation affecting the generalizability of findings.
- Tables 4 and 5 should explicitly note the direction of significant deviations from DRVs. Consider adding effect sizes where appropriate.
- Include additional considerations such as reporting bias, potential seasonal effects on vitamin D intake, and limitations of DRVs derived from broader populations.
- The explanation of abbreviations is missing below Table 1.

Author Response
The manuscript represents an important contribution to the field of nutrition and women’s health. The study provides valuable insight into the nutritional habits of perimenopausal women in Ireland and the relationships between nutrient intake and menopausal symptom severity. The methodology is appropriate, and the results are clearly presented. The manuscript is well-prepared and presents results of public health relevance. However, incorporation of the recommendations will enhance clarity, robustness, and impact of the work.
Thank you for your comments. The relevant changes have been addressed and the manuscript is now hopefully at a publishable standard. The authors would like to thank the reviewers for their very helpful feedback.
Recommendations and comments:
Comment 1: In the introduction it is stated that “It has also been suggested that having a high BMI may be positively correlated with a more symptomatic perimenopause [14]. Midlife obesity is also associated with reduced quality of life [15] and an increased risk of chronic disease [16,17].” However, I consider it important to complement this information with current research showing that what appears to be particularly risky in midlife is metabolically unhealthy obesity, which is associated with metabolic abnormalities and, in addition to a less favourable lipid profile, is often accompanied by elevated levels of uric acid and liver enzymes—factors typically linked to metabolic syndrome (g., “Differences in body composition between metabolically healthy and unhealthy midlife women with respect to obesity status”). It is also worth mentioning research that have found abdominal obesity to negatively influence the occurrence of menopausal symptoms in women of European origin („Association of menopausal symptoms with obesity in Slovak women“).
Response 1: These are helpful points and we have incorporated them into the introduction;
Line 60: ‘It has also been suggested that having a high BMI and central adiposity may be positively correlated with a more symptomatic perimenopause [14, 15]’. Midlife obesity is also associated with reduced quality of life [16], and an increased chronic disease risk [17,18]. Menopause is also associated with significant metabolic changes and a shift towards a more atherogenic lipid profile [19, 20].
- Luptáková, L., Siváková, D., Čerňanová, V., Cvíčelová, M., & Danková, Z. (2014). Association of menopausal symptoms with obesity in Slovak women. AR, 77(1), 57–66. https://doi.org/10.2478/anre-2014-0005.
- van Oortmerssen, J.A.E; Mulder, J.W.; Kavousi, M.; , Roeters van Lennep, J.E. Lipid metabolism in women: A review. Atherosclerosis 2025, 405, 119213. https://doi.org/10.1016/j.atherosclerosis.2025.119213.
- Vorobeľová, L.; Falbová, D.; Siváková, D. Differences in Body Composition Between Metabolically Healthy and Unhealthy Midlife Women With Respect to Obesity Status. AR 2021, 84, 59-71. https://doi.org/10.2478/anre-2021-0008.
Comment 2: „Age, physical activity, and marital status were included as covariates in the regression models, as they have previously been shown to be associated with menopausal symptom severity [40,41].“ However, the cited evidence is drawn exclusively from Chinese populations. Given that menopausal symptoms are known to be strongly influenced by ethnic and population-specific factors, it would be important to incorporate findings from studies conducted in European-origin populations as well. These studies have demonstrated that physical activity, obesity (BMI > 30, WHR > 0.89), alcohol intake, and socio-demographic factors—such as marital status and place of residence (town vs. village)—significantly affect menopausal symptomatology in midlife women. For example, menopausal status, BMI, residence, and alcohol consumption were identified as significant predictors of vaginal dryness. Moreover, psychological midlife symptoms were found to be primarily associated with physical health, lack of physical activity, use of oral contraceptives, alcohol consumption, and smoking in previous research (e.g., “The importance of female reproductive history on self-reported sleep quality, mood, and urogenital symptoms in midlife”; “The reproductive parameters, lifestyle and health factors in relation to physiological menopausal symptoms in Slovak women”). Some of these findings—particularly those regarding the influence of alcohol on menopausal symptoms—would be valuable to address in the discussion.
Response 2: Thank you for highlighting these relevant studies and for your comments. We have added additional references to the statistics section to include European data section [references 15 and 45] Lines 162-164;
Line 162: ‘Age, BMI, physical activity, and marital status were included as covariates in regression models as they have been shown previously to be associated with menopausal symptom severity [15,43-45]’.
We have added the following sentence to the paragraph on alcohol and caffeine;
Line 341: ‘Previous research in Slovakian midlife women has shown that alcohol consumption is associated with vaginal dryness and night sweats [45]’.
- Vorobeľová, L.; Falbová, D.; Candráková Čerňanová, V. The importance of female reproductive history on self-reported sleep quality, mood, and urogenital symptoms in midlife. Menopause 2023, 30, 1157-1166 https://doi.org/10.1097/GME.0000000000002277
Comment 3: The association between higher dietary vitamin E intake and elevated urogenital symptom scores is unusual. The Discussion should further explore potential mechanisms, confounding factors, or alternative explanations.
Response 3: Thank you for this. We have added to the discussion regarding confounding factors as no other research (or mechanism) at present supports this finding;
Line 389: ‘Conversely, the current study found an unexpected association between a higher intake of vitamin E and an increased urogenital sub-score, with a statistical model including co-variates accounting for 26% in the variance in symptom severity for this domain. However, this may be explained by the high rate of supplement use. Notably, 75% (n=47) of participants were taking supplements containing varying amounts of vitamin E including multivitamins, menopause specific supplements and evening primrose oil. Research has suggested an association between more severe menopausal symptoms are complementary and alternative medicine use [74]. Phytoestrogens and tocopherols are modulators of oestrogen receptors and have been suggested as a useful alternative to MHT in the management of menopausal symptoms [75], however there are no definitive guidelines on dosage.’
- Dehghan, M.; Kaviani, M.; Ebadollahi-Natanzi, A.; Bahrami, M.; Ghaffari, F. Use of Complementary and Alternative Medicine and Associations with Menopausal Symptoms in Postmenopausal Women: A Cross-Sectional Study. Front. Public Health 2022, 10, 947061. https://doi.org/10.3389/fpubh.2022.947061
Comment 4: The overrepresentation of highly educated participants should be acknowledged as a limitation affecting the generalizability of findings.
Response 4: Thank you. We have amended the text to reflect this comment;
Line 403: ‘The current study was limited through the use of a convenience sample. This resulted in a cohort with most participants (90%) having received a university level education, disproportionate to the national average of 65% of women educated to third level in Ireland [72] and as such results cannot be generalised. Future research utilising a more diverse population (e.g., women with different educational levels, socio-economic status, ethnicity etc.) is needed’.
Comment 5: Tables 4 and 5 should explicitly note the direction of significant deviations from DRVs. Consider adding effect sizes where appropriate.
Response 5: Thank you for this suggestion. We have now added a column to both tables 4 and 5 that includes the effect size for any significant results. Additionally, these values have been added to the text alongside the p values.
Comment 6: Include additional considerations such as reporting bias, potential seasonal effects on vitamin D intake, and limitations of DRVs derived from broader populations.
Response 6: Thank you for your helpful comments. We have added to this section to reflect your feedback;
Line 416: ‘However, self-reported data introduces the potential for reporting bias and estimated data is subject to errors which must be acknowledged [74]. Additionally, while DRVs are broad targets aimed at populations, they can overlook groups with distinct physiological or metabolic needs’.
Due to the Northern latitude in Ireland, subcutaneous vitamin D synthesis is limited. We have also added to the text to explain this;
Line 365: ‘Although seasonal variations in vitamin D were not accounted for in the current study, subcutaneous vitamin D synthesis is limited in Ireland due to its northern latitude (52-55°N) [65]’
Comment 7: The explanation of abbreviations is missing below Table 1.
Response 7: Thank you. Abbreviations for BMI and MHT have now been added below the table.
Round 2
Reviewer 2 Report
Comments and Suggestions for Authors
I agree with the proposed corrections, as they have significantly improved the overall quality of the article.